# Pemafibrate Improves Alanine Aminotransferase Levels Independently of Its Lipid-Lowering Effect

**Azuma Watanabe** [1],*, **Ryoko Horigome** [1], **Yumiko Nakatsuka** [1] **and Shuji Terai** [2]

1    Department of Gastroenterology, Kameda Daiichi Hospital, 2-5-22 Nishimachi, Konan-ku,
     Niigata City 9500165, Niigata, Japan; hrgm_ryonryon1127@nifty.com (R.H.); iyumi@ya2.so.ne.jp (Y.N.)
2    Division of Gastroenterology and Hepatology, Graduate School of Medical and Dental Sciences,
     Niigata University, 1-754 Asahimachidori, Niigata City 9518520, Niigata, Japan; terais@med.niigata-u.ac.jp
*    Correspondence: azuwata@ijn.or.jp; Tel.: +81-25-382-3111; Fax: +81-25-382-7311

**Abstract:** Aim: Non-alcoholic fatty liver disease (NAFLD) is the most common cause of chronic liver disease. Pemafibrate, a selective peroxisome-proliferator-activated receptor $\alpha$ modulator (SPPARM$\alpha$), has been reported to ameliorate liver function among patients with dyslipidemia. However, there are not many reports of the clinical effects of pemafibrate. This study aims to summarize the experience of using pemafibrate and analyze the effects on liver function in patients with dyslipidemia. Methods: One hundred twelve cases of hyperlipidemia receiving pemafibrate 0.2 mg/day were retrospectively enrolled in this study. Age, gender, BMI, complications, concomitant medications, serum parameters (TG, HDL-C, LDL-C, AST, ALT, $\gamma$GTP, ALP, platelets, M2BPGi, Cre, eGFR, HbA1c, blood glucose level at any time) were investigated and evaluated. Results: Pemafibrate administration significantly improved serum TG and HDL-C, but not in LDL-C. Serum AST, ALT, $\gamma$GTP, and ALP were also significantly improved. The fib-4 index, a liver fibrosis score, did not significantly change, but M2-BPGi, an index of fibrosis, significantly decreased. No correlation was observed between each lipid parameter and ALT, and ALT decreased independently of the lipid parameters. Conclusions: As we expected, pemafibrate demonstrated a lipid-improving effect without adversely affecting hepatic and renal functions. An unexpected finding was the decrease in ALT that was independent of lipid parameters.

**Keywords:** pemafibrate; Non-alcoholic fatty liver disease (NAFLD); alanine aminotransferase; M2-BPGi; dyslipidemia; liver fibrosis





## 1. Introduction

Dyslipidemia is defined as the presence of abnormal levels of lipids and lipoproteins in the bloodstream, typically characterized by elevated levels of total cholesterol (TC), low-density lipoprotein cholesterol (LDL-C), and triglycerides (TG), along with decreased levels of high-density lipoprotein cholesterol (HDL-C) [1].

Non-alcoholic fatty liver disease (NAFLD) is the most common cause of chronic liver disease, and its incidence is increasing [2]. NAFLD is frequently complicated by dyslipidemia and, in about 50% of cases, by hypertriglyceridemia (TG > 150 mg/dL) [3]. Diet, physical activity therapy, and the weight loss associated with them are the first choices for the treatment of NAFLD, but it is very difficult to achieve improvement because weight management is left to the motivation of a patient. In addition to weight loss, the next treatment that should be introduced is drug therapy. As a treatment method, Sodium-glucose cotransporter-2 (SGLT2) inhibitors, vitamin E, statins, Angiotensin-converting enzyme (ACE) inhibitors, and aldosterone receptor blockers have been proposed for Non-alcoholic steatohepatitis (NASH)/Non-alcoholic fatty liver disease (NAFLD) depending on complications, but there is currently no clear treatment method [4]. Fibrate, a peroxisome-proliferator-activated receptor (PPAR)$\alpha$ agonist, raises HDL-C and reduces TG. But it

is not selective and not a high-affinity ligand of PPARα. Bezafibrate activates not only PPARα but also PPAR γ/β and is considered a pan-PPAR agonist [5]. On the other hand, pemafibrate (Kowa Company, Nagoya, Japan), a selective peroxisome-proliferator-activated receptor (PPAR)-α modulator (SPPARMα), received the world's first approval in Japan as a therapeutic agent for dyslipidemia in 2018. It has a mechanism to lower TG more safely and efficiently by activating PPARα from a lower dose than conventional fibrates [6]. Pemafibrate has been reported to suppress hepatic fat deposition in a rodent model of NASH compared to fenofibrate [7]. Phase II trials have shown useful improvements not only in lipid profiles but also in hepatobiliary system parameters [8]. From these points, there is a report recommending pemafibrate in the treatment of NAFLD [4]. However, there are not many reports of the clinical effects of pemafibrate other than clinical trials. The purpose of this study is to summarize the experience of using pemafibrate and analyze what kind of patients are suitable for pemafibrate administration.

## 2. Methods

A retrospective observational study was conducted on dyslipidemia patients who received pemafibrate as outpatient treatment from April 2019 to April 2020. Cases were collected under the following conditions in the Department of Gastroenterology at our hospital (Kameda Daiichi Hospital): age, gender, Body mass index (BMI), complications, concomitant medications, serum parameters (TG, HDL-C, LDL-C, aspartate aminotransferase (AST), alanine aminotransferase (ALT), γ-glutamyl transpeptidase (γGTP), alkaline phosphatase (ALP), platelets, Mac-2 binding protein glycosylation isomer (M2BPGi), creatinine, eGFR (estimated glomerular filtration rate), HbA1c (hemoglobin A1c), blood glucose level at any time). These parameters were evaluated through non-fasting blood sampling, and the blood sampling time was unified as much as possible. Pre-administration data were obtained from the outpatient visit immediately before the administration of pemafibrate, and for post-administration data, the information from the last visit after the administration of pemafibrate was used. The present study was approved by the Ethical Committee of Kameda Daiichi Hospital (Institutional review board no. R3-2021, 28 April 2021) and consent to participate in this study was obtained using the opt-out method.

### 2.1. Liver Function Evaluation

The criteria for NAFLD were fat deposition on abdominal ultrasonography. The FIB-4 index was calculated to assess liver fibrosis [9]. FIB-4 index was calculated using the following formula: age (year) × AST (U/L)/platelet count (×$10^9$/L) × [ALT (U/L)]$^{1/2}$. Since serum ALT has been evaluated as a marker for the progression of liver fibrosis in NASH patients [10,11], the correlation between serum ALT and each parameter was examined.

### 2.2. Statistical Analysis

Each item value is expressed as mean ± standard deviation (SD) or %. Comparison before and after administration was performed using paired *t*-test, and the significance level was 5% on both sides. For correlation, the relationship between variables was evaluated using the Spearman correlation coefficient, and the significance level was set to 5%. For statistical analysis, Excel statistics and statistical analysis software EZR version 1.61 (Saitama, Japan; https://www.jichi.ac.jp/saitama-sct/SaitamaHP.files/statmedEN.html, accessed on 16 October 2023) were used.

## 3. Results

### 3.1. Baseline

This study included one hundred twelve sequential patients with hyperlipidemia receiving Pemafibrate 0.2 mg/day. The average administration period was 224.1 ± 83.6 days. An amount of 80% had liver disease and 63.4% were diagnosed with NAFLD, which was treated with ursodeoxycholic acid (UDCA) and vitamin E. The complication rates of lifestyle-related disease, hypertension and diabetes, were 48.2% and 38.4%, respectively,

and Sodium-glucose cotransporter-2 (SGLT2) inhibitors, Dipeptidyl peptidase-4 (DPP4) inhibitors, and metformin were administered as concomitant drugs. As therapeutic drugs for dyslipidemia, statins and ezetimibe were administered in 45.5% and 4.5%, respectively. There were no cases of concomitant use of Eicosapentaenoic acid (EPA) or Docosahexaenoic acid (DHA) preparations (Tables 1–3).

**Table 1.** Characteristics of patients (N = 112).

| | |
|---|---|
| Age (years) | 62.2 ± 13.7 (Mean ± SD) |
| Males/Females | 77/35 |
| BMI (kg/m$^2$) | 25.4 ± 4.2 (Mean ± SD) |
| Follow-up period (days) | 224.1 ± 83.6 (Mean ± SD) |

**Table 2.** Complications of patients (n).

| | |
|---|---|
| Liver disease | 90 (80.4%) |
| NAFLD | 71 (63.4%) |
| Hypertension | 54 (48.2%) |
| Diabetes mellitus | 43 (38.4%) |

**Table 3.** Concomitant medications of patients (n).

| | |
|---|---|
| Statin | 51 (45.5%) |
| Ezetimibe | 5 (4.5%) |
| EPA·DHA | 0 (0%) |
| SGLT2 inhibitor | 31 (27.7%) |
| DPP4 inhibitor | 28 (25.0%) |
| Metformin | 28 (25.0%) |
| Angiotensin II receptor blockers | 16 (14.3%) |
| UDCA | 30 (26.8%) |
| Vitamin E | 23 (20.5%) |

*3.2. Pre- and Post Treatment*

Significantly improved lipid parameters of TG and HDL-C were observed. There was no significant change in LDL-C. The hepatobiliary system parameters of aspartate aminotransferase (AST), alanine aminotransferase (ALT), and alkaline phosphatase (ALP) were significantly improved. The fib-4 index, which is a liver fibrosis score, did not significantly change. Body weight and platelets increased significantly. No significant changes were observed in renal and blood glucose parameters (Table 4).

**Table 4.** Changes in clinical parameters before and after pemafibrate therapy.

| Variables | Before | After | *p* Value |
|---|---|---|---|
| Weight (kg) | 68.3 ± 14.2 | 68.9 ± 14.1 | <0.05 |
| BMI (kg/m$^2$) | 25.4 ± 4.2 | 68.9 ± 14.1 | <0.05 |
| AST (IU/L) | 36.1 ± 32.1 | 27.8 ± 17.0 | 0.005 |
| ALT (IU/L) | 43.7 ± 43.8 | 24.0 ± 13.8 | <0.001 |
| γ-GPT (IU/L) | 93.8 ± 210.2 | 44.3 ± 129.3 | <0.001 |
| Platelet (10$^4$/μL) | 24.6 ± 7.2 | 26.9 ± 8.6 | <0.001 |
| M2BPGi | 0.9 ± 0.7 | 0.7 ± 0.5 | <0.05 |
| Triglyceride (mg/dL) | 234.6 ± 126.3 | 126.7 ± 68.4 | <0.001 |
| HDL-C (mg/dL) | 52.6 ± 12.9 | 55.7 ± 11.6 | 0.001 |
| LDL-C (mg/dL) | 118.4 ± 35.2 | 111.9 ± 28.0 | 0.06 |
| Cre (mg/dL) | 0.8 ± 0.2 | 0.8 ± 0.2 | 0.57 |
| eGFR (mL/min/1.73 m$^2$) | 73.0 ± 15.2 | 73.1 ± 15.5 | 0.70 |
| Glucose (mg/dL) | 138.7 ± 42.1 | 130.4 ± 38.1 | 0.15 |
| HbA1c (%) | 6.5 ± 1.0 | 6.4 ± 1.0 | 0.87 |

*3.3. ALT Correlation*

The correlation between ΔALT and the degree of changes in lipid parameters (ΔTG, ΔHDL-C, Δ LDL-C) and weight was examined. Correlation between ΔALT and ΔTG: Correlation coefficient r = 0.0319, *p* = 0.975; ALT and HDL-C: Correlation coefficient r = −0.173, *p* = 0.082; ΔALT and ΔLDL-C: Correlation coefficient r = 0.0196, *p* = 0.845; ΔALT and body weight: Correlation coefficient r = 0.0039, *p* = 0.845; and no correlation was observed between each lipid parameter and ΔALT, and body weight and ΔALT. ΔALT decreased independently of the lipid parameters and body weight (Figure 1).

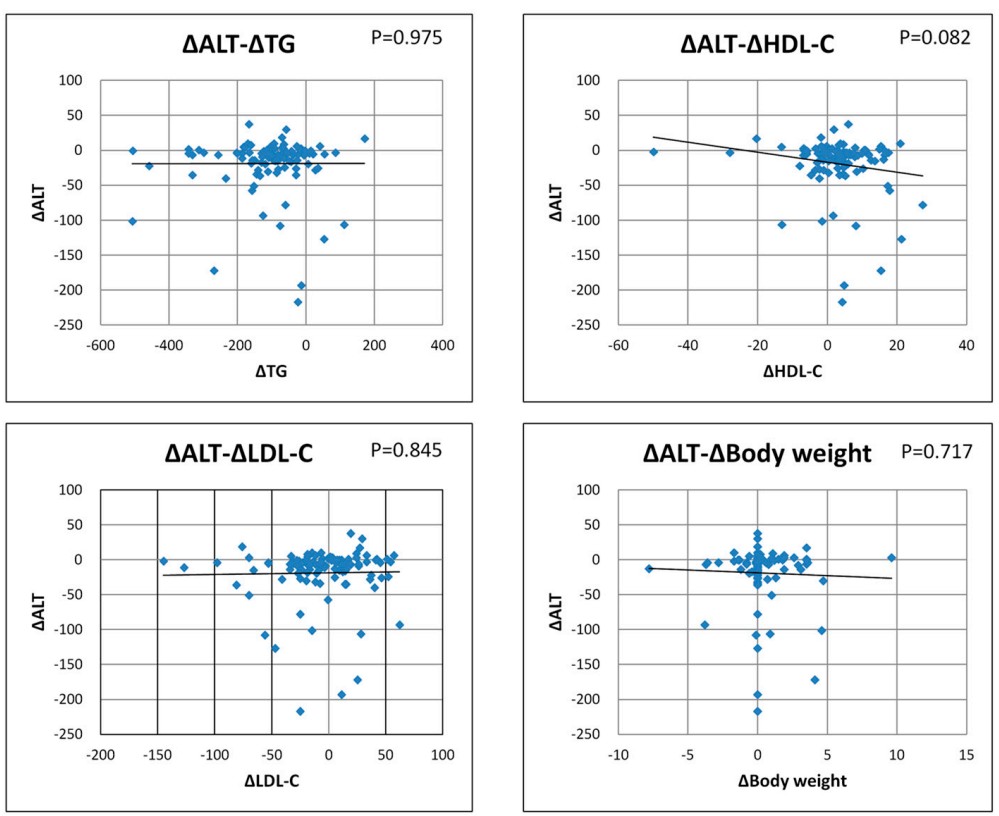

**Figure 1.** Association of ALT changes with lipid parameters and body weight.

## 4. Discussion

In this study, a significant improvement in lipid parameters and hepatobiliary parameters was observed in dyslipidemia patients who received pemafibrate. An amount of 64% of these cases were complicated by NAFLD, suggesting a link between NAFLD and hyperlipidemia. Hypertriglyceridemia and NAFLD are related diseases in metabolic syndrome, and NAFLD is often associated with hyperlipidemia. The complication rate in this study is almost the same as the previous report that showed that about 50% of patients with TG > 150 mg/dL were complicated by NAFLD [3]. Hypertriglyceridemia is also an exacerbating factor for cardiovascular events [12,13]. Cardiovascular disease is the most common cause of death in NAFLD patients [14]. Therefore, treatment intervention for lipid parameters is considered to be a necessary treatment for improving the long-term prognosis of NAFLD patients. In a Japanese phase II trial, the administration of pemafibrate reduced serum ALT levels in subjects with normal liver function [7]. There were significantly fewer adverse events associated with elevated hepatobiliary enzymes than in patients receiving fenofibrate. Patients with persistently abnormal aminotransferase levels are at high risk of NAFLD and development of liver fibrosis. It is important that the upper limit of ALT should be 30 U/L, because high ALT is associated with increased liver-related mortality [15]. In this study, AST and ALT are decreased, which is in agreement with the previous findings. Several clinical studies have reported the clinical effects of pemafibrate

in patients with NAFLD, but they are limited [16–18]. Since serum ALT has been evaluated as a marker for the progression of liver fibrosis in NASH patients [10,11], we performed a correlation analysis between the degree of ALT changes and lipid parameters in this study. As a result, it was confirmed that ALT changed independently of lipid parameters. In addition, ALT was not correlated with body weight. Pemafibrate decreased collagen 1α1 and TNFα mRNA expression in the liver with NASH model mice [7]. The improvement in liver fibrosis and inflammation through pemafibrate treatment might reduce the serum levels of ALT in the present study. In a pemafibrate phase 2 study in NAFLD patients, not only with hypertriglyceridemia but also with non-hypertriglyceridemia, pemafibrate therapy significant reduced in serum ALT and liver stiffness [19]. Therefore, these findings suggest that it might directly improve liver fibrosis and alleviate inflammation in the liver, not via triglycerides' lowering effects. This is consistent with the improvement in ALT levels independently of its lipid-lowering effects in the present study. The present study also showed the reduction in the biliary enzymes, γGTP and ALP. Fibrate activates PPARα and micellizes hydrophobic bile acids via the upregulation of the expression of multidrug resistance gene 3 (mdr3), a related transporter for secretion of biliary phospholipid in bile duct membranes [20,21]. According to the result, fibrate may protect hepatic cells and the bile duct epithelium. Therefore, it has been reported that the efficacy of pemafibrate treatment adds to ursodeoxycholic acid in primary biliary cholangitis patients with dyslipidemia [22,23].

This is a new finding that has never been reported before. PPARα knockout mice develop liver inflammation, steatosis, and carcinogenesis [24,25]. Therefore, PPARα is key to improving fatty liver. Pemafibrate is a drug that promotes mitochondrial β-oxidation in hepatocytes and lowers lipid parameters, especially TG, by activating the nuclear receptor PPARα [26]. Honda et al. reported that pemafibrate reduced hepatic fat, hepatocyte ballooning, and hepatocyte inflammation/fibrosis. The number of macrophages and tumor necrosis factor (TNF-α) messenger RNA (mRNA) expression decreased, and steatosis grade was significantly lower in the liver of mice treated with pemafibrate [7]. Sakai et al. also reported that pemafibrate suppressed hepatic inflammation. An increased hepatic lipid droplet number and reduced size were observed in NASH model mice [27]. Even in LDL knockout pigs that do not exhibit hyperglyceridemia, pemafibrate administration suppresses vasculitis [28]. This is thought to be a direct effect on blood vessels. The ALT-lowering effect of pemafibrate in this study may also be contributed by a direct anti-inflammatory effect on the liver and may have been caused by a different mechanism from the serum-TG-lowering pathway. PPARα is an important factor for improving fatty liver, but conventional PPARα agonists, that is, fibrates such as fenofibrate and bezafibrate, adversely affect liver function and have little advantage in treating patients with NAFLD [6]. Pemafibrate, which is more selective for PPARα than fenofibrate/bezafibrate, may have had beneficial effects on NAFLD reported in mouse models [7]. This high selectivity may help reduce the occurrence of side-effects such as liver and kidney damage. In this study, statins were prescribed in half of the cases during the observation period of 1 year or more, but there were no significant changes in renal markers. From the above points, it is considered that the risk of adverse effects on the kidneys is low.

Our results suggest that pemafibrate not only significantly reduces triglyceride levels but also improves serum ALT levels independently of lipid parameters. There is potential for pemafibrate to be considered as a future treatment for NAFLD.

## 5. Conclusions

Pemafibrate lowers triglycerides without inducing hepatic and renal dysfunction. Furthermore, it reduces ALT levels independently of lipid parameters. Pemafibrate may have beneficial effects on NAFLD. It is a safe and efficient medication for treating dyslipidemia and NAFLD.

## 6. Limitation

There are several limitations in this study.

1. This was a single-facility, retrospective observational study.
2. The pemafibrate administration period was not unified.
3. The control group was not set.
4. No histopathological evaluation of the liver was performed after administration.
5. There are quite a lot of concomitant medications, and pemafibrate alone could not be evaluated. In particular, SGLT2 inhibitors and concomitant drugs such as statins and ezetimibe may greatly affect the effects of NAFLD.

**Author Contributions:** A.W. performed conceptualization, data acquisition and analysis, writing of the manuscript; R.H. performed data acquisition; Y.N. performed data acquisition; S.T. performed project administration and manuscript editing. All authors have read and agreed to the published version of the manuscript.

**Funding:** S.T. receives research funding from Kowa company.

**Institutional Review Board Statement:** The present study was approved by the Ethical Committee of Kameda Daiichi Hospital (Institutional review board no. R3-2021, 28 April 2021).

**Informed Consent Statement:** The present study was consent to participate in the study was obtained using the opt-out method.

**Data Availability Statement:** The data that support the findings of this study are available from the corresponding author upon reasonable request.

**Acknowledgments:** Vladimir Bilim; Department of Urology, Kameda Daiichi Hospital, Niigata, Japan.

**Conflicts of Interest:** S.T. receives research funding from Kowa company. The rest authors declare no conflict of interest.

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
