# Peer review of "Pemafibrate Improves Alanine Aminotransferase Levels Independently of Its Lipid-Lowering Effect"

_livers, doi:10.3390/livers3040038_

Round 1
Reviewer 1 Report
In their study, the authors have retrospectively analyzed 112 patients with hyperlipidemia treated with pemafibrate to assess its effects on liver function. Pemafibrate significantly improved serum TGs and HDL-C Liver enzymes including AST, ALT, γGTP, and ALP also showed significant improvements.
Notably, the decrease in ALT levels occurred independently of changes in lipid parameters. Overall, pemafibrate demonstrated favorable lipid-modulating effects without harming hepatic or renal functions, and it unexpectedly lowered ALT levels. This finding raises several important points that the authors should consider with the following comments:
1. Regarding the visual elements in the manuscript, it is apparent that the figures and tables need significant improvement. The current presentation lacks clarity and effectiveness in conveying the data.
2. The author should prioritize clarity for readers by providing full explanations of abbreviations used in the manuscript. Examples; ALT, UDCA, EPA, etc..
3. The author has to provide an explanation of how the results contribute to the advancement of our understanding of dyslipidemia management and its relevance to potential NAFLD treatments.
4. Providing a more comprehensive background on lipids will help readers better understand the context and significance of the study's findings.
5. It is crucial for the author to explore deeper into the unexpected finding of ALT improvement without a clear correlation with lipid parameters. A thorough analysis and discussion of potential factors or mechanisms underlying this phenomenon would significantly enrich the paper, providing a more comprehensive understanding for readers and potentially contributing to the broader field of dyslipidemia management and NAFLD treatment.
6. While recent studies have examined the relationship between lipids and ALT levels, the author should prioritize explaining the practical significance of ALT in the context of NAFLD. Specifically, the author should elaborate on how ALT serves as a valuable biomarker for diagnosing and monitoring NAFLD, shedding light on its clinical utility.
Author Response
Thank you for your time and effort on the manuscript. We hope that thanks to your suggestions, the manuscript has significantly improved.
- Regarding the visual elements in the manuscript, it is apparent that the figures and tables need significant improvement. The current presentation lacks clarity and effectiveness in conveying the data.
*** We appreciate your feedback. We have made significant improvements to our tables and figures to enhance clarity and comprehension.
- The author should prioritize clarity for readers by providing full explanations of abbreviations used in the manuscript. Examples; ALT, UDCA, EPA, etc..
***Thank you for this valuable comment. Abbreviations are now defined upon their first use in the manuscript.
- The author has to provide an explanation of how the results contribute to the advancement of our understanding of dyslipidemia management and its relevance to potential NAFLD treatments.
***We thank you for this valuable input. We have extensively revised the discussion and expanded the conclusion to elucidate how our results can contribute to the advancement of dyslipidemia management.
- Providing a more comprehensive background on lipids will help readers better understand the context and significance of the study’s findings.
***Your insights have been highly constructive. Additional information has been incorporated into the introduction to address the problem mentioned above.
- It is crucial for the author to explore deeper into the unexpected finding of ALT improvement without a clear correlation with lipid parameters. A thorough analysis and discussion of potential factors or mechanisms underlying this phenomenon would significantly enrich the paper, providing a more comprehensive understanding for readers and potentially contributing to the broader field of dyslipidemia management and NAFLD treatment.
***Your insights have been highly constructive. we have thoroughly reworked the discussion and expanded the conclusion to improve the overall flow of the manuscript.
- While recent studies have examined the relationship between lipids and ALT levels, the author should prioritize explaining the practical significance of ALT in the context of NAFLD. Specifically, the author should elaborate on how ALT serves as a valuable biomarker for diagnosing and monitoring NAFLD, shedding light on its clinical utility.
***Thank you for your comment. In my reply to points 3 and 5, I have also already tried to address this comment. The current discussion aims to provide a comprehensive response to these concerns.

Reviewer 2 Report
The communication submitted for expertise is a study which aims to list experiments carried out on the use of pemafibrate and analyze the effects on liver function in patients suffering from dyslipidemia. The study is carried out on one hundred and twelve cases of hyperlipidemia receiving pemafibrate in a dose of 0.2 mg/day. The size of the sample ensures good homogeneity and correlation of the results. The authors show that pemafibrate has a lipid effect without altering liver and kidney functions. On the other hand, the discovery is unexpected with regard to the increase in ALT independent of lipid parameters. This should be better developed.
Revisions suggestions;
- "conclusion" can be improve
- add more recent references
- Table 1 "Males / Females" ?
- Complete reference 17
- reformat Table 2

Author Response
Thank you very much for your comment. We appreciate the time and effort you've put in.
- "conclusion" can be improve
***Thank you for your comment. We changed the conclusion. We hope that in the present form, it has become more clear.
- add more recent references
***Thank you for your suggestion. We added more recent references as you recommended.
- Table 1 "Males / Females" ?
- reformat Table 2
***Thank you for your comment. We thoroughly edited all tables and made corrections to them.
- Complete reference 17
***Thank you for your comment. As I mentioned above, we added more recent references as you recommended and checked them for consistency.
